# Beyond Hypoglossal Hype: Social Media Perspectives on the Inspire Upper Airway Stimulation System

**DOI:** 10.3390/healthcare11233082

**Published:** 2023-12-01

**Authors:** Nicholas A. Rossi, Bridget A. Vories, Samuel E. Razmi, Nishat A. Momin, Zachary S. Burgess, Harold S. Pine, Sepehr Shabani, Rizwana Sultana, Brian J. McKinnon

**Affiliations:** 1Department of Otolaryngology, University of Texas Medical Branch, Galveston, TX 77555, USA; namomin@utmb.edu (N.A.M.); zsburges@utmb.edu (Z.S.B.); hspine@utmb.edu (H.S.P.); seshaban@utmb.edu (S.S.); brmckinn@utmb.edu (B.J.M.); 2School of Medicine, University of Texas Medical Branch, Galveston, TX 77555, USA; bavories@utmb.edu; 3School of Medicine, Texas A&M College of Medicine, EnMed Initiative, Houston, TX 77030, USA; samrazmi@tamu.edu; 4Department of Internal Medicine, Division of Pulmonary Critical Care & Sleep Medicine, University of Texas Medical Branch, Galveston, TX 77555, USA; risultan@utmb.edu

**Keywords:** sleep apnea, obstructive, electric stimulation therapy, social media, health communication, patient education as topic, telemedicine, implantable neurostimulators, hypoglossal nerve

## Abstract

In the landscape of sleep surgery, the Inspire^®^ Upper Airway Stimulation (UAS) device has gained prominence as an increasingly popular treatment option for obstructive sleep apnea, prompting significant discourse across social media platforms. This study explores the social media narrative of the UAS device, particularly the nature of multimedia content, author demographics, and audience engagement on Instagram, Facebook, and TikTok. Our analysis encompassed 423 public posts, revealing images (67.4%) and videos (28.1%) as the dominant content types, with over a third of posts authored by physicians. A notable 40% of posts were advertisements, whereas patient experiences comprised 34.5%. TikTok, although presenting a smaller sample size, showed a substantially higher engagement rate, with posts averaging 152.9 likes, compared with Instagram and Facebook at 32.7 and 41.2 likes, respectively. The findings underscore the need for otolaryngologists and healthcare professionals to provide clear, evidence-based information on digital platforms. Given social media’s expanding role in healthcare, medical professionals must foster digital literacy and safeguard the accuracy of health information online. In this study, we concluded that maintaining an evidence-based, transparent digital dialogue for medical innovations such as the UAS device necessitates collaborative efforts among physicians, health institutions, and technology companies.

## 1. Introduction

Upper airway stimulation (UAS) has been a breakthrough technique in the domain of sleep surgery for the treatment of obstructive sleep apnea (OSA) [1,2,3,4,5]. Once a novel approach, it has rapidly ascended in popularity, transforming from an innovative technique to a mainstay in sleep surgery clinical practice. UAS, also known as hypoglossal nerve stimulation, represents a novel therapeutic approach to obstructive sleep apnea, particularly in patients who struggle with continuous positive airway pressure (CPAP) therapy. This treatment involves the surgical implantation of a device that stimulates the hypoglossal nerve, which controls tongue movements. By targeting this nerve, UAS prevents the tongue from collapsing backward during sleep, thereby maintaining an open airway. The device’s activation is timed with the patient’s breathing cycle, ensuring that airway obstruction is minimized during sleep. This approach offers a less invasive alternative to traditional OSA treatments, focusing on improving sleep quality and reducing apnea events.

The rapid rise of this procedure and its recognition by both the medical community and the lay public can be seen in the influence of social media on disseminating healthcare information. With the contemporary patient being more connected than ever, healthcare information exchange, particularly via social media, has reached unprecedented levels [6,7,8,9]. However, this increasing reliance on social media as a primary source of health-related information brings forth a surfeit of concerns [10]. Despite its growing acceptance in the medical community, the procedure remains in its nascent stages, with long-term outcomes still under investigation [1,4,11]. As the boundary between authentic medical advice and influencer-led endorsements becomes increasingly blurred on social platforms, the responsibility of physicians to convey accurate, evidence-based information intensifies [10,12,13].

Social media’s integration into the fabric of modern healthcare communication presents both opportunities and challenges [7,14]. While social media platforms offer a vast, accessible arena for healthcare discourse, they also create avenues for misinformation, partial information, and biased narratives. This phenomenon is especially pronounced in the case of new medical technologies like UAS, where public understanding is still evolving. Consequently, there is a critical gap in understanding the nature and impact of social media narratives on UAS. Our research aims to fill this gap by systematically analyzing the portrayal of UAS on key social media platforms. By doing so, we seek to provide insights into the current digital discourse surrounding UAS, identify prevalent themes and narratives, and assess their potential impact on patient perceptions and decision-making. This study is particularly pertinent given the increasing reliance of patients on social media for healthcare information and healthcare professionals’ need to navigate and influence these digital conversations effectively [15,16,17,18]. Thus, our investigation not only illuminates the current state of social media discussions on UAS but also can guide future strategies for accurate and ethical health communication in the digital age.

Recent years have seen a burgeoning interest in analyzing social media content to gauge public perception of various medical procedures. Previous studies have analyzed posts related to various surgical conditions and procedures, such as sinus surgery, pediatric scoliosis, and hip arthroscopy [19,20,21]. Similarly, research by other authors in areas such as pediatric tonsillectomy, rhinoplasty, and cochlear implants has emphasized the role of social media in shaping patient expectations and experiences [22,23,24], revealing a trend towards patient-authored narratives and educational content. This trend toward patient-centric narratives and the educational role of social media aligns with our investigation into the Inspire^®^ UAS device, suggesting a common theme across different medical domains. OSA remains a relatively underexplored area in otolaryngology social media research.

In this study, we investigated the portrayal and reception of UAS on three of the most popular social media platforms, Instagram, TikTok, and Facebook, highlighting the influence of digital discourse on patient understanding. In examining the digital narratives surrounding UAS technology, we aimed to emphasize the physician’s potential role in ensuring accurate and balanced online health communication. Moreover, we addressed potential gaps or misconceptions about UAS prevalent on social media, underscoring the imperative of an authentic and comprehensive representation of emerging medical technologies in the digital domain.

## 2. Materials and Methods

This investigation employed a mixed-methods research design, incorporating both qualitative and quantitative data collection methods. We systematically gathered public data from three prominent social media platforms: Instagram, Facebook, and TikTok. Quantitative data included engagement metrics such as likes, shares, and comments, while qualitative data encompassed thematic analyses of post content and authorship demographics. The research pivoted around five predefined search terms: #inspiresleep, #inspiresleepapnea, #inspiresleepapneaimplant, #hypoglossalnervestimulator, and #lifewithinspire. These search terms were neither sponsored nor designed by industry but were chosen by the authors given that these hashtags yielded the highest number of posts at the time of investigation. Data was collected over a defined span from April 2018 to April 2023 by four independent investigators. Posts that bore no relevance to the Inspire^®^ UAS device or were written in non-English languages were excluded. Since this research hinges on publicly accessible social media content, it was deemed exempt from Institutional Review Board (IRB) approval per the extant guidelines of the IRB at our affiliated institution.

### 2.1. Multimedia Classification

Posts sourced from these platforms were systematically categorized based on their intrinsic multimedia elements: image, video, or text. Instagram accommodates either image or video uploads, TikTok specializes exclusively in video content, whereas Facebook allows users to post images, videos, or text.

### 2.2. Authorship Demographics

We thoroughly examined publicly available user profiles to determine the authorship of each post (Table 1). This process included assessing profile information, historical content, and any affiliations or disclosures provided within the account. This analysis enabled us to categorize authors into groups such as Inspire^®^’s official representatives, individual patients, family members of patients, practicing physicians, and other relevant categories. In cases where a patient created a post, classification was based on the user’s self-identification and the content’s context, indicating personal experiences or testimonials related to UAS. For posts potentially serving marketing purposes, we looked for indicators such as promotional language, links to commercial sites, or explicit affiliations with the Inspire^®^ company (Golden Valley, MN, USA) or other commercial entities. This methodology, while relying on information made publicly available by users, provided a framework to categorize each post’s origin accurately and assess the nature of its content.

### 2.3. Subject Categorization

The content elicited from these platforms underwent thematic segregation and was delineated into four overarching paradigms: advertisements, educational posts, patient experience, and media coverage. These subjects were defined as:**Advertisement**: Strategically oriented posts promoting the Inspire^®^ device, affiliated medical procedures, or ancillary resources.**Educational**: Scholarly content delving deep into the device’s mechanism, indications, therapeutic potential, or expansive knowledge pertinent to the Inspire^®^ Sleep Apnea Hypoglossal Nerve Stimulator.**Patient Experience**: In-depth chronicles delineating subjective encounters with the Inspire^®^ device, encompassing both the procedure and post-procedural experiences.**Media Coverage**: Analytical content highlighting the portrayal of or discourse on the Inspire^®^ modality within the broader media echelons, encapsulating news, documentaries, and similar channels.

### 2.4. Engagement Metrics

Engagement metrics, primarily the number of ‘likes’, were used to indicate audience interaction and post popularity. The ‘likes’ count, being a direct and quantifiable measure of engagement, offered a straightforward method to compare the impact of posts across different platforms. Although ‘shares’ and ‘comments’ were also considered, these metrics were not uniformly available or quantifiable across all platforms, limiting their utility in our analysis. For instance, while Facebook and Instagram display the number of shares a post receives, TikTok does not. Thus, our engagement analysis focused predominantly on the ‘likes’ metric, supplemented by qualitative observations of comments where available.

### 2.5. Data Collection Standards

We utilized a rigorous, standardized data collection guideline to ensure consistency and reliability in our data collection process (see Appendix A). This guideline included detailed procedures for identifying relevant posts, classifying multimedia elements, determining authorship, and categorizing post subjects. It also provided clear definitions and examples for each category and variable, aiming to minimize subjective interpretation and enhance the objectivity of the data collected. This standardized approach was crucial to maintaining the integrity of our mixed-methods research methodology.

## 3. Results

This study included 423 social media posts related to UAS and the Inspire^®^ UAS device extracted from three platforms (Table 2). Instagram was predominant, yielding 308 posts. This was followed by Facebook and TikTok with 92 and 23 posts, respectively.

### 3.1. Multimedia Classification

Content dissemination based on media type indicated images as the prevalent format, accounting for 67.4% (285/423) of the posts. Videos constituted 28.1% (119/423) of posts. Text-only posts were exclusively found on Facebook, making up 4.5% (19/423) of the total.

### 3.2. Authorship Demographics

Physicians were the predominant authors in this cohort, contributing 32.2% (136/423) of the posts (Figure 1), with 28.6% (121/423) of the posts originating from the Inspire^®^ company itself. Patients additionally represented 25.3% (107/423) of posts. Other contributors included non-physician healthcare providers, academic institutions, professional organizations, and media outlets.

### 3.3. Subject Categorization

Advertisements formed 40.0% (169/423) of the posts (Figure 2). Patient experiences made up 34.5% (146/423) of the content. Educational posts constituted 24.6% (104/423) of posts. Out of the Inspire^®^ company’s 121 posts, 79.3% (96/121) of posts were classified as advertisements, 9.1% (11/121) were educational posts, and 7.4% (9/121) were lifestyle-oriented posts.

### 3.4. Engagement Metrics

Engagement, as gauged by mean likes per post, portrayed TikTok as the platform with the highest user interaction, averaging 152.9 likes per post (Figure 3). By contrast, Instagram and Facebook demonstrated mean likes of 32.7 and 41.2 per post, respectively. Overall, the mean likes across all platforms stood at 41.1. TikTok posts demonstrated a significantly greater mean number of likes than posts from either Facebook or Instagram (*p* < 0.05). Posts written by patients garnered 133.7 likes per post, the highest of any author. By contrast, posts authored by physicians had 24.1 likes per post, and those made by the Inspire^®^ company averaged 27.5 likes per post.

### 3.5. Hashtag Analysis

Posts tagged with #hypoglossalnervestimulator predominantly originated from formal entities such as the Inspire^®^ company, physicians, or academic institutions, with an absence of this hashtag among patient posts. Furthermore, this hashtag garnered fewer likes compared to others. Multiple hashtag utilization within a single post was a common observation. The hashtag #inspiresleep was especially potent, resulting in the highest retrieval of posts related to the study’s focus. Notably, 94.1% (96/102) of posts utilizing the hashtag #lifewithinspire were made by the Inspire^®^ company.

## 4. Discussion

The Inspire^®^ Upper Airway Stimulation (UAS) device has been a focal point of discussion in sleep apnea and sleep surgery literature, as hypoglossal nerve stimulation has gained traction as a popular option for surgical management of OSA [25,26,27]. The device has been commonly advertised on platforms such as television, radio, and social media, the latter of which physicians have assumed a proactive role, with 32.2% of all posts related to the UAS device being written by physicians. This substantial physician engagement is a deviation from other otolaryngologic procedures and diagnoses such as tympanostomy tubes, cochlear implants, and laryngectomy, where physicians’ online participation has been significantly less [24,28,29]. This finding may suggest a growing recognition among the medical community about the benefits of the Inspire^®^ UAS device and a desire to spread awareness to the public of its potential in sleep medicine [1,2,30].

In this study, TikTok greatly eclipsed the other platforms in user engagement with an average of 152.9 likes per post, compared with 32.7 and 41.2 likes per post for Instagram and Facebook, respectively. This finding underscores TikTok’s potential burgeoning influence on healthcare conversations [31,32,33]. Further highlighting this trend is that patient-authored narratives garnered the highest average likes across posts. Such profound engagement with firsthand patient experiences may suggest an evolving appetite among the audience for genuine, personal insights into medical interventions while emphasizing the increasing role of newer social media platforms in shaping medical narratives. However, it is important to approach these findings with caution. TikTok’s significant lead in engagement metrics may be influenced by its relatively low volume of posts in our study, which could amplify the impact of individual posts on the platform’s average engagement figures. This fact necessitates careful consideration of the data, as a smaller sample size might not fully represent broader user interactions and may lead to overestimations of engagement levels.

Our study findings regarding the digital discourse surrounding the Inspire^®^ Upper Airway Stimulation (UAS) device aligned with and expanded upon the existing literature examining social media’s role in healthcare communication [6,17,34,35,36]. Our mixed-methods approach, combining both quantitative and qualitative data collection, provided a comprehensive understanding of the online narrative. Moreover, applying a rigorous, standardized data collection guideline also contributed to the robustness of our findings by ensuring the consistency and reliability of our analyses. The significant engagement of both physicians and patients in the social media discussion about UAS reflects trends observed in similar studies. For instance, prior analyses of social media narratives concerning other otolaryngologic procedures such as rhinoplasty, tympanostomy tubes, cochlear implants, and tonsillectomy revealed a pattern of diverse patient perspectives, mirroring our findings [22,23,29]. However, our study distinguishes itself by focusing on an emerging technology in sleep surgery, thereby contributing unique insights into how nascent medical innovations are discussed and perceived online. Furthermore, the predominance of advertisements in our study resonates with Moffatt et al. and Lahaye et al., who also reported a significant commercial presence in social media discussions on laryngectomy and new-age rhinology devices [28,37]. This observation underscores the increasing role of commercial interests in shaping public perceptions of medical technologies, a trend that necessitates a critical approach to evaluating online health information.

The high level of engagement on newer platforms like TikTok, as noted in our study, reflects a broader shift in digital health communication strategies [38,39]. Studies by Ramkumar et al. on shoulder and elbow surgery, cellular therapy, and joint arthroplasty using social media also highlighted the platform-specific dynamics of patient engagement and information dissemination [40,41,42]. Our research contributes to this discourse by emphasizing the significant impact such platforms have on shaping patient understanding and expectations, especially for novel treatments like UAS. Moreover, similar to Hairston and Haeberele’s studies, which analyzed parental perspectives on pediatric tonsillectomy and patient perceptions of hip arthroscopy [20,23], our research highlights the evolving nature of patient narratives on social media. These narratives, as evidenced in our study, often focus on personal experiences and insights, providing a valuable, albeit subjective, view of patient journeys and satisfaction with medical interventions like UAS. Considering these parallels and distinctions in the existing literature, our study underscores healthcare professionals’ need to engage proactively in digital spaces [43,44]. By contributing accurate, evidence-based information, physicians can help balance the narratives shaped by commercial interests and patient anecdotes. This approach is vital in ensuring that patients receive a comprehensive view of new medical technologies, aiding in informed decision-making.

While the active engagement of patients and physicians on social media underscores the increasing universal accessibility of healthcare discussions, ensuring the accuracy and integrity of disseminated information can be challenging [16,45,46]. The predominant role of advertisements and patient narratives emphasizes the need for a balanced portrayal, encompassing both the merits and potential risks of the UAS device. The distinct dynamics observed across platforms, particularly TikTok’s ascendant influence, reflect a generalized shift in modern healthcare communication strategies. Physicians, as primary stakeholders and trusted voices, are encouraged to guide these narratives, ensuring that they remain both informative and evidence-based while resonating with the evolving digital milieu of their audience [8,47,48].

While several studies have examined various aspects of otolaryngology on social media [19,23,24,28,29,37,49], the presence of UAS on social media has not been thoroughly studied. In a 2021 study, Xiao et al. evaluated the usefulness and informativeness of YouTube videos for patients concerning UAS for OSA [50]. They found that many of these videos lacked comprehensive and quality information on the subject. Although YouTube is a popular source of information for patients, YouTube videos on UAS were considered insufficient in many content areas. Given the increasing demand for information among both the medical and lay public regarding UAS, there must be more high-quality and comprehensive resources for patients seeking knowledge on this topic.

In the age of information ubiquity, medical data sources play a pivotal role in determining its legitimacy and consequent patient actions. Given the substantial traction of advertisements and patient narratives on the Inspire^®^ UAS device on social media, there is a pressing need to address digital trust and literacy [51,52]. Patients, when met with an influx of diverse information, must be able to discern evidence-based content from anecdotal accounts or commercial promotions. Likewise, physicians, as stewards of accurate health communication, should be equipped with the knowledge to evaluate and curate content in the vast digital expanse [53,54]. The challenge transcends content generation and helps foster a digitally literate audience that approaches online medical data with informed skepticism and critical engagement. This goal necessitates a multipronged approach, where physicians, medical institutions, and tech platforms collaborate to fortify the integrity of online medical narratives, ensuring they resonate truthfully within the rapidly evolving digital ecosystem.

The digital landscape presents both opportunities and challenges for sleep medicine practitioners. Given the potent influence of platforms like TikTok, sleep medicine specialists are encouraged to embrace these platforms proactively, promoting the emergence of authoritative, evidence-based narratives. A set of best practices could assist physicians in creating transparent content, distinguishing between patient anecdotes, research-backed findings, and commercial promotions. Collaborations with tech platforms could be explored to devise algorithms prioritizing verified medical information, especially for topics with substantial patient impact like the Inspire^®^ UAS device [45,55]. Furthermore, continuous medical education programs must integrate modules on digital literacy and communication, equipping physicians with the tools to not only disseminate but also critically assess online content [56,57,58,59]. As digital health communication evolves, the sleep medicine community must lead with informed engagement, safeguarding the integrity of the information ecosystem in the digital age.

The significant presence of advertisements in our dataset, constituting 40% of the content, offers a critical perspective on the nature of information dissemination about medical technologies like UAS. The dominance of promotional material raises concerns about potentially skewing public perception, primarily when these advertisements emphasize the benefits of UAS while downplaying its risks or limitations [60,61]. This trend has implications for patient expectations and decision-making, highlighting the need for healthcare professionals to provide a balanced view encompassing both the advantages and potential drawbacks of such treatments. While our study focuses on a surgical device produced in the United States, the global reach of social media platforms suggests that these findings have wider relevance [52,62,63,64,65]. The interpretation and impact of social media content on UAS can vary significantly across different countries, influenced by cultural norms, healthcare systems, and regulatory environments. For example, in countries with restricted access to UAS or differing healthcare policies, the portrayal of UAS on social media might affect patient advocacy and demand differently than in the United States. Developing strategies that recognize diverse cultural and regulatory landscapes can ensure the accessibility of social media content on medical technologies as well as accuracy, balance, and cultural sensitivity [66,67,68]. As UAS continues to gain traction globally, understanding its portrayal on social media becomes imperative for shaping effective and ethical healthcare communication strategies worldwide.

This investigation is not without limitations. A primary constraint is that the methodology involved convenience sampling of public social media posts, which is inherently non-randomized and must be interpreted in the context of the specific posts examined. Our focus on the platforms Instagram, Facebook, and TikTok may narrow the study’s generalizability, especially given the global popularity of other platforms [69,70,71]. Future research should broaden the scope to encompass a wider range of social media platforms such as Snapchat, LinkedIn, or Twitter, offering an additional lens into the UAS discourse. Employing technologies like artificial intelligence for data analysis, which was beyond the scope of this study, might further enhance the depth and breadth of our findings. Moreover, by surveying only public posts, we missed insights from private accounts or posts limited to specific user networks, a choice often made for privacy reasons. Additionally, the influence of industry stakeholders, who may act as de facto influencers, represents a further limitation. Their promotional activities might skew the representation and perception of UAS on social media, potentially overemphasizing benefits while underreporting risks or adverse outcomes. The subjective nature of post categorization, such as determining sentiment or intent, is another potential limitation. A different research group might derive varied conclusions due to individual biases or interpretations. Study findings can be influenced by the ever-changing nature of social media platforms, their algorithms, and user behaviors, as with any study involving dynamic platforms. Therefore, the clinical application of these insights should be approached judiciously and balanced with evidence-based practices and individual patient considerations.

## 5. Conclusions

The prominence of the Inspire^®^ Upper Airway Stimulation (UAS) device on social media underscores the vital interplay between medical innovation and the digital realm, particularly on platforms like TikTok. This study highlights the dynamic nature of healthcare communication in the digital age, revealing a narrative landscape that blends patient experiences, educational content, and a significant proportion of advertisements. Such findings accentuate sleep medicine specialists’ need to engage actively in these digital spaces with transparent, evidence-based narratives. This approach is crucial to counterbalance promotional content, fostering informed patient decision-making in an era where social media increasingly influences health-related perceptions.

Our investigation, while comprehensive, acknowledges the limitations of its scope and methodological approach, suggesting directions for future research. Expanding the analysis to more social media platforms and employing advanced analytical tools can offer deeper insights into the global discourse surrounding medical technologies like UAS. As the field of sleep medicine continues to evolve alongside digital communication channels, a collaborative and strategic approach is essential. By embracing digital literacy and working closely with tech platforms, healthcare professionals can ensure the integrity and effectiveness of the healthcare information ecosystem, catering to a globally diverse audience.

## Figures and Tables

**Figure 1 healthcare-11-03082-f001:**
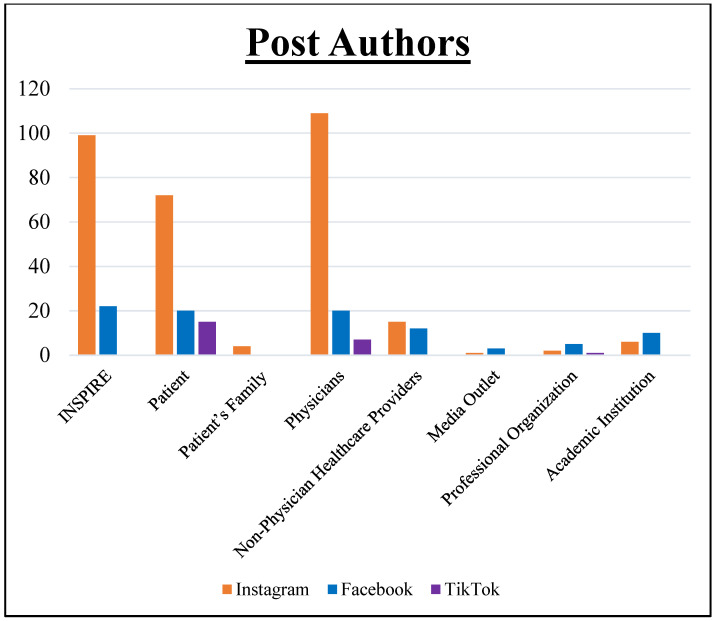
Social media post authors: Bar graph illustrating the distribution of social media post authors across platforms, highlighting a prominent presence of physicians, especially on Instagram.

**Figure 2 healthcare-11-03082-f002:**
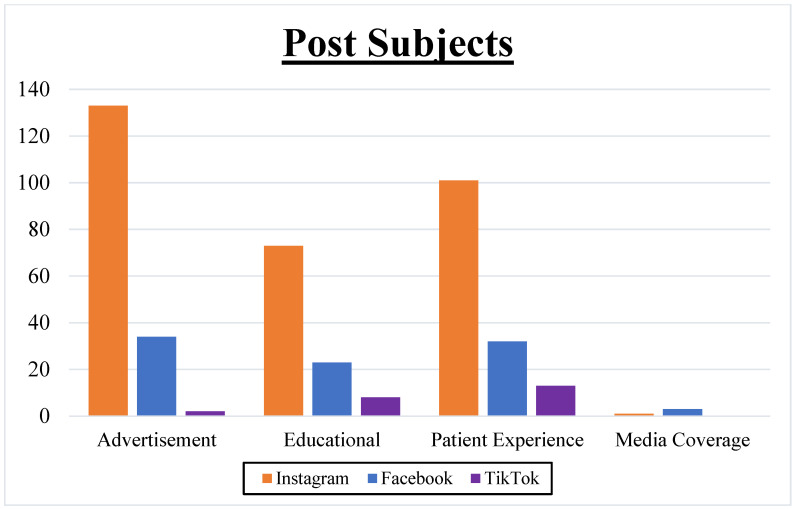
Social media post subjects: Graphical representation comparing subjects of social media posts across platforms, showcasing a high prevalence of advertisement and patient experience posts.

**Figure 3 healthcare-11-03082-f003:**
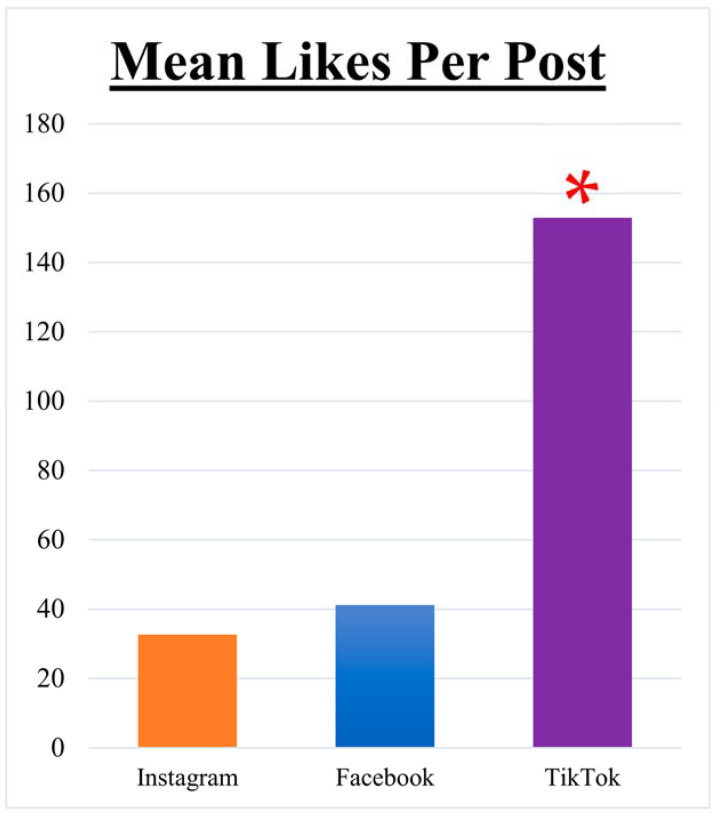
Mean likes per post: Bar graph depicting the Mean Likes Per Post for Inspire^®^ Sleep Apnea content. Note that the red asterisk (*) indicates a significantly greater mean number of likes per post on TikTok than on Instagram or Facebook.

**Table 1 healthcare-11-03082-t001:** Classification and descriptions of social media post topics related to Inspire^®^ Sleep Apnea.

Inspire^®^ Social Media Post Topics	
Subject	Description
** *Inspire^®^* **	Official posts from the company, often involving new research findings, advancements in therapy, patient success stories, or announcements about events such as webinars and conferences.
** *Patient* **	Personal testimonials and experiences. These range from the initial diagnosis of sleep apnea and the decision to use Inspire^®^ to the journey of adjustments and eventual improvement in sleep quality.
** *Patient’s Family* **	Observations and reflections from family members, noting the difference in their loved one’s energy levels, mood, and overall well-being post-treatment.
** *Physicians* **	Clinical insights and observations. This encompasses discussions about the efficacy of Inspire^®^, comparison to other treatments, or even patient recovery anecdotes.
** *Non-Physician Healthcare Providers* **	These posts offer a unique perspective from healthcare providers who are not physicians but still interact with Inspire^®^ patients. They discuss post-operative care, common questions they receive, or general feedback on the device.
** *Media Outlet* **	Posts highlighting the appearance of Inspire^®^ in popular media. This includes articles, documentaries, interviews with patients or medical experts, and any other media feature.
** *Professional Organization* **	Discussions or endorsements from reputed medical associations or groups. These posts focus on the technical aspects of the therapy, research collaborations, or recommendations for usage.
** *Academic Institution* **	Posts from research institutions studying the effectiveness of Inspire^®^. These range from clinical trial announcements, research findings, or even student experiences working with Inspire^®^ patients.

**Table 2 healthcare-11-03082-t002:** Distribution of Inspire^®^ social media data by platform, type, author, subject, and popularity.

Inspire^®^ Social Media Data					
		n (%)			
**Platform**		Instagram	Facebook	TikTok	**Total**
	Included Posts	308	92	23	**423**

**Type**					
	Image	221	64	0	**285 (67.4)**
	Video	87	9	23	**119 (28.1)**
	Text	0	19	0	**19 (4.5)**

**Author**					
	Inspire^®^	99	22	0	**121 (28.6)**
	Patient	72	20	15	**107 (25.3)**
	Patient’s Family	4	0	0	**4 (0.9)**
	Physicians	109	20	7	**136 (32.2)**
	Non-Physician Healthcare Providers	15	12	0	**27 (6.4)**
	Media Outlet	1	3	0	**4 (0.9)**
	Professional Organization	2	5	1	**8 (1.9)**
	Academic Institution	6	10	0	**16 (3.8)**

**Subject**					
	Advertisement	133	34	2	**169 (40.0)**
	Educational	73	23	8	**104 (24.6)**
	Patient Experience	101	32	13	**146 (34.5)**
	Media Coverage	1	3	0	**4 (0.9)**

**Popularity**	Mean Likes Per Post	32.7	41.2	152.9	**41.1**

## Data Availability

The datasets generated and analyzed during the current study are not publicly available as they were not initially prepared for distribution. However, they are available from the corresponding author upon reasonable request.

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
