# Peer review of "Beyond Hypoglossal Hype: Social Media Perspectives on the Inspire Upper Airway Stimulation System"

_healthcare, 2023, doi:10.3390/healthcare11233082_

Round 1
Reviewer 1 Report
Comments and Suggestions for Authors
The manuscript entitled “Beyond Hypoglossal Hype: Social Media Perspectives on the Inspire Upper Airway Stimulation System” was interesting. The following comments can help the authors to improve it:
1- Please choose appropriate keywords using the MeSH terms.
2- The rational for conducting the research is not clear.
3- The introduction section needs to be expanded to include the related literature and the gap in the existing knowledge.
4- The methodology of the research is not clear and a mixture of indicators/variables has been considered to extract and report data.
5- It is not clear how the results can be useful in other countries. In particular, according to Figure 1, most posts were related to advertisement.
6- Please follow the journal style for referencing.
Author Response
The authors thank the reviewer for their comments. Please see the attachment for detailed author response.

Reviewer 2 Report
Comments and Suggestions for Authors
Thank you for inviting me to review the manuscript. The research methodology related to this project didn’t follow scientific approaches. Authors mentioned that you collected data from public social media platforms. How authors measured interactions via social media? How authors confirmed the patient status and thier use for the UAS?! What about marketing posts? The methodology as well result are not scientifically sound. Major improvements and revision are needed for this manuscript.
All the best
Comments on the Quality of English LanguageFine
Author Response

(The authors gave the same response as above.)

Reviewer 3 Report
Comments and Suggestions for Authors
Dear Editors,
Thank you very much for allowing me to review this article. The authors have done an interesting and courageous work by analyzing a medical aspect in the context of social networks. Although the impact of this type of article remains to be seen, innovative approaches like this one should be analyzed with a future perspective because of their potential. Therefore, this article may be of interest. However, I would like to propose some comments and suggestions to the authors:
I appreciate the concise presentation, which enhances readability. The introduction aptly outlines the two topics discussed in the paper. However, I suggest that the authors provide further justification for their research.
The authors should provide more detailed information on specific aspects in the methodology section. For instance, they mention that the search terms used in their study were those with the highest number of posts. However, it is unclear how the authors determined this and what previous research was conducted to identify these search terms. The previous study used to identify the search terms may have introduced biases depending on the methodology and criteria used. Therefore, it would benefit the readers to provide additional information on the process used to determine the search terms.
The methodology description is mixed up with the results. For instance, the second point of section 2.3 talks about the posts obtained, which looks like a quantitative result rather than a methodology. Similarly, point 2.4 talks about identifying topics, which is a qualitative result, not a description of the methodology. Additionally, Table 1 is mentioned long before it is shown, which makes it difficult to understand. Finally, section 2.1 is missing, but this could be a typographical mistake.
The study is defined as qualitative by the authors. However, the results presented are primarily quantitative. It gives the impression that the study combines qualitative and quantitative results. While the methodology includes the identification of themes, which is a qualitative aspect, the results section is based on numerical data, which is quantitative. I recommend that the authors define their methodology more clearly. It is important to note that readers who prefer one methodology or the other may feel uncomfortable if the methodology used is not clearly defined.
I recommend that the authors remove the gray background in the figures, which would allow the figures to be cleaner.
One of the study's most important limitations is the potential selection bias. The social networks chosen for the period and the topics (or hashtags) introduce an unavoidable bias. The potential biases of the methodology do not invalidate the study since it is also understood to be exploratory. However, it would be helpful if the authors clearly described this limitation and, if possible, assessed the external validity of their results against it.
In summary, the authors should review certain aspects of their article, particularly the organization of the methodology and the results. It would be helpful to provide a more precise justification for the study and contextualize the potential selection bias that may have arisen from the search methodology used. Despite these issues, the article has particular potential to explore a relatively unknown terrain and can serve as a valuable resource for similar studies.
Author Response

(The authors gave the same response as above.)

Round 2
Reviewer 1 Report
Comments and Suggestions for Authors
I appreciate the authors for their time and efforts to revise the manuscript. It has now improved significantly.
Reviewer 2 Report
Comments and Suggestions for Authors
The authors did well in revising the manuscript. I have no further comments or suggestions to add.